# Dynamic Physical Activity Recommendation Delivered through a Mobile Fitness App: A Deep Learning Approach

**Subramaniyaswamy Vairavasundaram [1,*], Vijayakumar Varadarajan [2,3,*] , Deepthi Srinivasan [1], Varshini Balaganesh [1], Srijith Bharadwaj Damerla [1], Bhuvaneswari Swaminathan [1] and Logesh Ravi [4,5]**

1 School of Computing, SASTRA Deemed University, Thanjavur 613401, India; skdeepthi123@gmail.com (D.S.); varshinibalaganesh005@gmail.com (V.B.); srijithdamerla2001@gmail.com (S.B.D.); s.bhuvana@sastra.ac.in (B.S.)
2 School of Computer Science and Engineering, University of New South Wales, Sydney, NSW 2052, Australia
3 School of NUOVOS, Ajeenkya DY Patil University, Pune 412105, India
4 SENSE, Vellore Institute of Technology, Chennai 600127, India; logeshphd@gmail.com
5 Data Engineering Research Group, Vellore Institute of Technology, Chennai 600127, India
* Correspondence: vsubramaniyaswamy@gmail.com (S.V.); v.varadarajan@unsw.edu.au (V.V.)

**Abstract:** Regular physical activity has a positive impact on our physical and mental health. Adhering to a fixed physical activity regimen is essential for good health and mental wellbeing. Today, fitness trackers and smartphone applications are used to promote physical activity. These applications use step counts recorded by accelerometers to estimate physical activity. In this research, we performed a two-level clustering on a dataset based on individuals' physical and physiological features, as well as past daily activity patterns. The proposed model exploits the user data with partial or complete features. To include the user with partial features, we trained the proposed model with the data of users who possess exclusive features. Additionally, we classified the users into several clusters to produce more accurate results for the users. This enables the proposed system to provide data-driven and personalized activity planning recommendations every day. A personalized physical activity plan is generated on the basis of hourly patterns for users according to their adherence and past recommended activity plans. Customization of activity plans can be achieved according to the user's historical activity habits and current activity objective, as well as the likelihood of sticking to the plan. The proposed physical activity recommendation system was evaluated in real time, and the results demonstrated the improved performance over existing baselines.

**Keywords:** data-driven; machine learning; tracking physical fitness; personalized recommendation system; mobile apps; walking step count; fitness activity

**MSC:** 00A06

## 1. Introduction

People in today's world have to contend with hectic and active lifestyles on a daily basis. Therefore, one of society's primary challenges is encouraging people to adopt or continue leading healthy lifestyles to reduce their risk of developing chronic conditions. As a key driver for safeguarding good health from a preventive point of view, aligned with the pursuit of Sustainable Development Goal (SDG) 3 "good health and wellbeing", it is, therefore, essential to engage and motivate citizens with healthy activities that are tailored to their interests, as this will serve as a key driver in the process. This is one of the reasons why health recommendation systems have recently become a trend in research, particularly in food and physical activity recommendation. A regular exercise regimen strongly affects our physical and mental health when we make it an everyday habit [1]. In the United States, only 21% of adults above 21 years of age meet the Centers for Disease Control and Prevention (CDC) guidelines regarding physical activity [2]. Despite being aware of these

physical activity standards, people do not regularly meet their goals for various reasons, including a lack of gyms and unfamiliarity with the importance of physical activity. With the recent emergence of fitness trackers and other wearable devices, such as smartwatches and wristbands, sensors capture accurate data about individuals' physical activity [3,4] and transmit it through smartphone applications. In these studies, the authors demonstrated the tracking of steps by smartphone apps and wearable devices. Another study [5] proved that these devices accurately tracked step counts. Other studies recorded accelerometer data to accurately classify body movements such as lying, walking, and sit-to-stand [6].

Even though such technologies have gained popularity, their impact on health behaviors is still unclear. While such devices may result in initial adoption and behavior change, their eventual abandonment is likely high. A study [7] interviewed users of popular fitness gadgets and revealed why users no longer use their devices or applications. Even though existing technologies provide motivation and reminders of previous activity levels, they do not provide strategies to reach physical activity targets. Participants stated that they wanted their gadget to assist with an adaptive plan technique according to their unexpected plans in their official and personal lives. These users' requirements provided the inspiration for our presented work. In addition, personalized treatments have been proven effective in trials to improve computational-powered physical behavior activities [8]. Hence, we followed a data-driven, tailored advisor system [9] in this work to enable physical activity planning [10].

Considering walking as a realistic and accessible method of promoting general health, this study uses step counts to measure physical activity. By giving users of fitness trackers hourly strategies, we help them reach a daily goal on the basis of their past actions and other factors. According to research on existing adaptive therapies [9], most efforts are devoted to adjusting daily targets weekly according to earlier successes. One study [9] used smartphone software to subdivide daily goals into more manageable chunks during the day. The segmentation algorithm used by the authors was not disclosed in their work.

When the user is less likely to achieve their daily goal at specific points during the day, the proposed method adjusts the activity plan and the daily objective following the prediction to ensure that the daily goal is achieved successfully. Despite anticipating that a random user will reach their everyday objective, the system does not change the goal in response to the identified prediction. The authors of [11] developed a system that offers users certain activities on the basis of past attendance to similar activities (e.g., modest walks within the specific region). The proposed method of this current work strives to meet both criteria because users not only need assistance in detecting behavioral changes, but they also require detailed instructions on how to make those changes.

Due to improvements in technology such as digital watches and mobile phones that can more precisely forecast data, collecting real-time statistics with high precision has become simple. Physical activity or step count numbers and physiological characteristics were included in this dataset. The physiological aspects included the user's height, weight, gender, body mass index (BMI), and age. The physical activity dataset contained each individual's hourly step count value for 42 days. Contextual factors were a combination of physiological and physical characteristics such as the number of training sessions, average workout time, and average calories burned per session.

We utilized step count as our fundamental measurement since walking is the most recommended and everyday activity of many people and is also practicable [10]. We use this information to suggest activities to users on the basis of their hourly activity patterns. The novelty of the presented work lies in the exploration of the long short-term memory (LSTM) algorithm to forecast everyday goals on an hourly basis. The developed Android-based mobile application tracks the user's hourly step count and estimates whether the participant has achieved their everyday target. If they fall short, then the app is structured to shift the target to successive hours so as to achieve the daily goal.

As the mobile application provides hourly planning, it allows users to track their progress toward daily goals. In addition, consumers receive reminders if they are on track.

This gives the user a better view of the difference between their usual walking activity and their daily goal-based walking activity, allowing them to promptly adapt to their behavior. The key contributions of the proposed work are as follows:

1.  Our system utilizes historical activity data to determine typical activity patterns using two-level clustering. Daily activity plans are developed and adjusted on the basis of these patterns throughout the day.
2.  According to the current levels of activity achievement and a user's typical activity pattern, our system predicts whether a user will reach their daily goal. Using this prediction, the system can tailor notifications, present content, and possibly modify the current plan.
3.  Our system can generate realistic adaptations of a user's original plan when the probability of them achieving their activity target falls below or above predefined threshold levels.

The remainder of the paper is organized as follows: Section 2 elaborates on existing research related to deep learning techniques that can be used to enhance adherence to a physical activity regimen. Section 3 highlights the significance of the project's clustering, classification, and predictive models. The experimental results are presented in Section 4, along with their evaluation. Lastly, conclusions, along with future work directions, are presented in Section 5.

## 2. Related Works

In recent years, recommendation systems have been put to use in fields such as e-commerce, tourism, and multimedia streaming. These are all areas in which it is essential to personalize users' experience on the basis of their interactions; therefore, these fields have found recommendation systems to be particularly useful. Recent developments in recommendation systems have also focused on wellbeing; however, existing solutions have only been designed considering a single aspect of wellbeing at a time, such as a healthy diet or an active lifestyle.

Physical activity can improve our health exponentially and reduce the risk of diseases such as type 2 diabetes, cancer, and cardiovascular disease. It has immediate and long-term health benefits. Regular exercise can improve our quality of life. A previous study developed a similar project that provides an hourly activity plan based on users' past activity patterns and provides personalized recommendations for each day. They used hierarchical clustering and predictive methods to construct an adaptive, data-driven advisor for increasing physical activity [9]. Since it was a health-related project, using machine learning techniques gave less accuracy when compared with the deep learning approach.

In [12], Abhaya Kumar Sahoo, Chittaranjan Pradhan, and Himansu Das experimented with different machine learning algorithms. The same experiments were then performed with deep learning techniques. On comparing the results of the above experiments, it was found that deep learning algorithms outperformed machine learning algorithms on data related to health, specifically diabetes. Convolution neural network-based deep learning provided high accuracy when compared to other algorithms. The authors of [13] used bidirectional LSTM network learning to monitor machine health. In this project, a real-life wear test tool was tested using this model, which predicted the actual tool wear from the sensory data; it was concluded that the LSTM model has given the best result.

In [14], on the basis of step count data, machine learning techniques such as support vector machines and logistic regression were applied to predict relapsing exercise users. The discontinuation prediction score (DiPS) of these classifiers was calculated using the run-in period data of the user, e.g., their daily step counts. In addition, the system employed the DiPS score as an early warning score to predict when the user would relapse from regular exercise. According to the report, DiPS offered highly accurate and robust predictions and high specificity and sensitivity. In [15], using ensemble machine learning techniques, the authors demonstrated the high accuracy of DiPS prediction over other approaches, including logistic regression and support vector machine. In [16], a mobile application that

partitions users' daily targets into various segments until the end of the day was developed, but the algorithms used to perform this segregation were not specified.

The popularity of wearable activity tracker devices can be attributed to several factors, including self-determination [17], self-awareness, motivation, tracking progress, and remaining informed [18]. It has been demonstrated that wearable activity tracker devices can increase one's motivation to exercise through various constructs [19]. Some concepts are connected to people's interactions [19]. Others relate to the exercise control features and the data management features as potential avenues for data analysis, collection, progress updates [19], or constructive feedback [17]. These are all aspects of exercise. It has been established that daily sociability, psychological factors such as high extroversion levels, and behavioral factors such as large network size are important modulators of the physical activity implication in young populations [20]. Providing highly active participants with personalized feedback helps facilitate positive emotional responses. Subjects with low activity levels are more likely to experience negative emotional responses but also have a better chance of developing positive coping mechanisms [21].

Activity trackers have demonstrated the potential to increase physical activity; however, the effects on weight loss continue to be inconsistent [22]. In addition, the use of tracker devices to record daily activity (calories) has been linked to the possibility of initiating, maintaining, or exacerbating eating disorders [23]. Previous research [24–28] has focused on the adoption of technology products in different age ranges; however, research on the differences in the adoption of wearable activity tracker devices between generations is limited. For instance, research on the use of wearable technology by adolescents revealed conflicting results [29,30]. In addition, there are still many unanswered questions concerning the differences in the use of tracking devices between men and women. It has been demonstrated that more women than men have participated in research studies investigating the effectiveness of wearable activity devices when incorporated into all-encompassing weight loss programs [31]. It is also essential to keep in mind that the functionality of wearable activity tracker devices is not guaranteed to be accurate [32]. The need to adapt the type of device to the characteristics of the users has also been discussed, and it has been seen that it is important for long-term use to facilitate the user experience in terms of functionality, aesthetics, and physical design [33]. The need to adapt the type of device to the characteristics of the users has also been discussed. Increasing the amount of evidence that is available on the use of tracker devices to record daily activity in order to encourage healthy habits is recommended in light of the contentious data regarding the effectiveness of such devices.

This research states that physical activity and daily exercise are necessary for a fit and healthy lifestyle. We propose a model based on a deep learning approach to predict and provide personalized health advice.

## 3. Proposed Methodology—Dynamic Physical Activity Recommendation

The whole process, from recognizing the user activity groups to developing the physical activity advisor, is split into two parts, as shown in Figure 1. For the generation of a new activity chart, the machine learning and recommendation phases are described under two scenarios: offline computations and online computations. The subsections below describe the dataset utilized, outline the preprocessing approaches, and propose two computation methods.

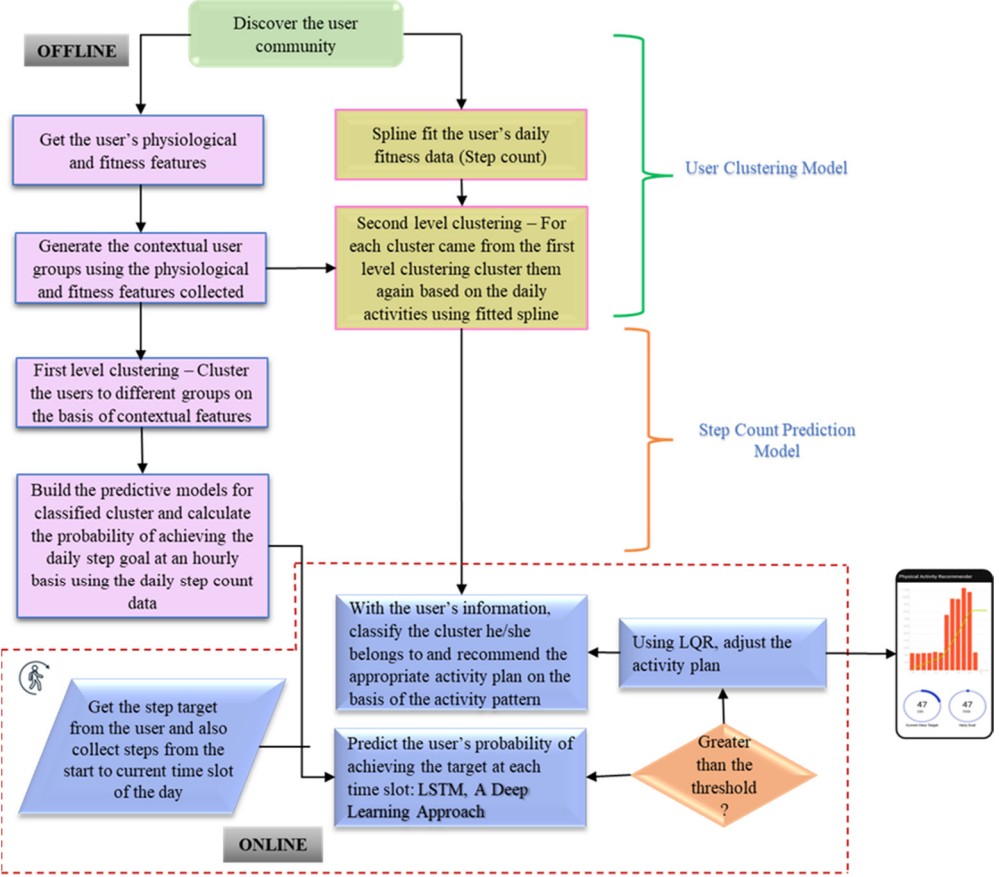

**Figure 1.** Flow chart of dynamic physical activity recommendation for fitness app.

### 3.1. Physical Activity Pattern Data

On the basis of the physical activity patterns of 533 participants, user data were separated into physical and physiological features. Physiological features included gender, age, height, weight, and BMI. There were six physical features of workouts recorded: average time spent in a workout, standard deviation of time spent in a workout, workout per rate, number of sessions, average number of calories burned, and standard deviation of the number of calories burned. The data were still unprocessed, thereby eliminating inconsistencies and impurities. These features helped to determine the cluster to which the user belonged. The physical features of users were based on step counts and burned calories. Fitness details for users were based on their historical step count over many days. The data on daily step counts for the 533 participants consisted of an average of 530 observations. Accordingly, the hourly calories a person burned and the total calories burned per day were calculated. Participants' historical data were used to determine their adherence to physical activity. The purpose of the large dataset was to provide an accurate result by determining how closely a person adhered to their physical activity. When the person did not follow the prescribed chart, the recommendation was given on the basis of their tracked walking steps. The definitions of the physiological and physical features considered for this study are listed in Table 1.

These features were used to determine the adherence to the physical activity routine, as well as to personalize recommendations. Furthermore, the user's daily and hourly step counts, the calories burnt, and the physical features were entirely calculated. The sample data for physical features are given in Table 2, while sample data for physiological features are given in Table 3. These data might be inconsistent due to their healthcare characteristics. Thus, preprocessing was required to make the data consistent by removing impurities, missing values, and outliers, allowing the data to be used for further computations.

**Table 1.** Definition of physiological and physical features.

| Physiological Features | Physical Features |
|---|---|
| 1. Gender: gender of the user<br>2. Age: age given by the user<br>3. Weight: weight given initially by the user<br>4. Height: height provided by the user<br>5. BMI: body mass index calculated using the user's height and weight | 1. Workout par rate: ratio of the number of times a user has a workout session to the number of days data are recorded<br>2. Average workout time: average time user spends on their workout<br>3. Standard deviation of workout time: standard deviation for the workout time<br>4. Average calories burnt: average amount of calories burnt by user<br>5. Standard deviation of calories burnt: standard deviation for calories burnt by user<br>6. Number of sessions: total number of sessions taken up by user |

**Table 2.** Physical features of users.

| ID | Gender | Age | Height | Weight | BMI |
|---|---|---|---|---|---|
| 1001 | Female | 23 | 1.6 | 52.0 | 20.31 |
| 1002 | Male | 19 | 1.8 | 80.0 | 24.69 |
| 1003 | Male | 20 | 1.83 | 66.0 | 19.70 |
| 1004 | Male | 25.0 | 1.78 | 78.0 | 24.61 |

**Table 3.** Physiological features of users (WT—workout time, CB—calories burnt, SD—standard deviation).

| W/Rate | Average WT | SD of WT | Average CB | SD of CB | Number of Sessions |
|---|---|---|---|---|---|
| 0.7 | 1.47 | 0.57 | 304.0 | 131.36 | 21 |
| 0.8 | 1.33 | 0.38 | 276.87 | 4.90 | 24 |
| 0.86 | 1.42 | 0.5 | 302.53 | 121.59 | 26 |
| 0.73 | 1.45 | 0.54 | 310.72 | 131.11 | 22 |

*3.2. Data Preprocessing*

The data were processed suitably for further use in the machine learning models. The accelerometer data for 533 individuals, 290 of whom were men and 243 of whom were women, were utilized for this work. The historical data were formatted, cleared, and merged. The necessary features were extracted using the dimensionality reduction technique [10]. The user activity patterns were generated on the basis of these features. The preprocessed data were free of all the missing values, data inconsistencies, and outliers. The event-driven approach was used for analyzing the step counts and calorie data [34]. The persistent analytical data were stored for the online processes of clustering and new activity chart generation [6]. A two-level clustering was performed to identify the users' contextual group and give them a personalized activity chart for the rest of the day. An analytical data store was created to store the user's persistent activity data and identify the cluster into which a new user would fall. Finally, the preprocessed data were applied for successive steps (e.g., data clustering and classification). For learning model purposes, the data were split into a 60:20:20 ratio for training, validation, and testing, respectively.

*3.3. Offline Computations*

In this mode of computation, the physical activity advisor application contained the inbuilt information-gathering phase. Instead, the history data were used to recognize the user population to determine their physical and physiological features, on the basis of which the contextual user groups were clustered. Then, the process of fitting user step count data, building predictive models, and second level clustering took place in sequential order.

**Discovering the user community:** Initially, all kinds of user populations were identified including those adherent to physical activity, as well as moderately active and

nonadherent people. Among these categories, people who required motivation and who were not adherent in nature to the physical activity plan were the target audience for this work.

**Getting the user's physical and fitness features:** The specified user population provided the physical and fitness characteristics. Each user's physical factors, such as height, weight, BMI, age, and gender, as well as fitness characteristics, such as step count data and calories burned data, were gathered hourly. This information was preprocessed and utilized to create contextual user groups [35]. Impurities accounted for around 30% of the total and were eliminated. The empty values were interpolated [36]. This information was utilized to efficiently generate contextual characteristics.

**Generating the contextual features:** The particular collected physical and physiological features from each user were employed to create the contextual user groupings. These contextual features collectively included each customer's physical and fitness features for 2 months. These features could be used for first-level clustering. The in-text feature acted as the foundation for further offline calculations. The contextual features for the 533 users were employed to identify the contextual customer groups.

### 3.3.1. First-Level Clustering

The initial level of clustering was employed to identify different contextual end-user groups. The accumulated and merged in-text features received were used as input to the first-level clustering. The output of first-level clustering divided the overall discovered end-user population into many clusters according to similar contextual features. We used hierarchical agglomerative clustering [37] to cluster users into different in-text groups. This clustering method is popular in splitting the given data structured on their likeness. This algorithm takes each object to be a singleton cluster. Since the contextual features were divided using the user's data similarity, we used hierarchical agglomerative clustering.

### 3.3.2. Hierarchical Agglomerative Clustering

Hierarchical agglomerative clustering, also known as AGNES, clusters data by considering each object to be a singleton object [38]. This set of rules works in a bottom-up approach. The two most similar clusters based on the generated features using this algorithm are combined into a new bigger cluster at the end of each step. To measure the similarity between the objects, agglomerative clustering most commonly uses the Manhattan distance or the Euclidean distance. A specific function called dist () in this inbuilt clustering algorithm is used to calculate the clusters' similarity [39]. Using the specified distance measure, i.e., Jacobian distance measure or Euclidean distance measure (Equation (1)), this function calculates the similarity between the rows of the data matrix. Another function in agglomerative clustering called linkage () which is also an inbuilt function given in Equation (2), gets the value from the dist () and groups the objects into pairs on the basis of their similarity. Here, we used a single linkage to cluster the data into several groups. An example of a dendrogram representing the first-level clustering is shown in Figure 2.

$$\text{Euclidean distance, } d = \sqrt{\sum_{i=1}^{n}(x_i - y_i)^2}. \tag{1}$$

$$\text{Agglomerative clustering, } D(x, y) = \min_{x \in X, y \in Y} d\, x, y. \tag{2}$$

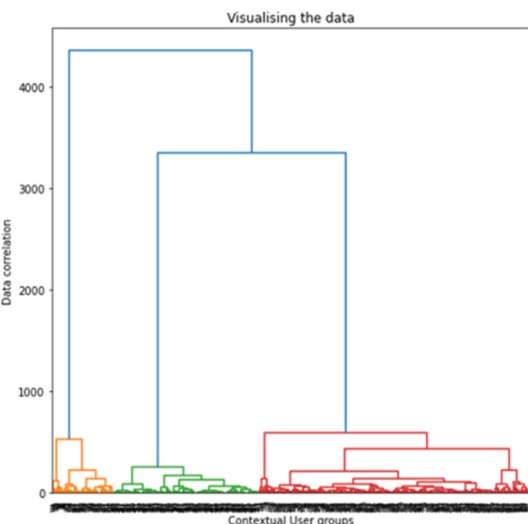

**Figure 2.** Dendrogram representation of first-level clustering.

The silhouette index determines the number of clusters that would be perfect for the process. The silhouette index gives a score for the number of clusters that can be formed ranging from 1 to *n*, as mentioned by the user. The highest score for the corresponding *n* is taken as the number of clusters and given to the agglomerative clustering algorithm to split the dataset accordingly. This algorithm gives the score for the number of clusters using the mean intra-cluster distance and the mean nearest-cluster distance for each data sample. The silhouette coefficient is also defined using specific criteria [12]. The values range between 1 and −1. A slide toward a negative value indicates that the cluster was assigned with the wrong data point. Thus, eventually, the value of that cluster decreases. A positive value indicates that the sample was posted to the right cluster, whereby a value of 1 indicates that the sample was accurately assigned to the specific cluster. This method gives the average silhouette coefficient over all samples [40]. The range for the silhouette coefficient is given in Equations (3) and (4).

$$\text{Range for defining Silhouette coefficient} = 2 <= \text{n\_labels} <= \text{n\_samples} - 1. \tag{3}$$

$$\text{Silhouette coefficient} => (b - a)/\max(a, b). \tag{4}$$

Thus, the number of clusters into which the data can be split is defined using the silhouette index and is given to the agglomerative clustering to be split accordingly. Then, the clustered data are provided for further prediction and classification. Here, we compared *n* values from 2 to 7, and the obtained silhouette index was higher for *n* = 3, with a value of 0.81.

### 3.3.3. User Classification: Random Forest

To identify the category of each new user in terms of the contextual user group, the data were trained using classification models such as decision tree, support vector machine (SVM), and random forest. On the basis of the initial accuracy of models using this user dataset, random forest responded with high accuracy in the experimental scenario. Thus, to predict the user's activity and properly assign the contextual user group [41], the primary role of this classification was to determine the contextual user group for new users. The result obtained from the first-level clustering was used to train the random forest classification model. This classification model aimed to classify the user with either partial or complete data. As random forest is a bagging ensemble algorithm, it can handle large user datasets efficiently. Specifically, random forest can be trained quickly with high-dimensional data over a decision tree algorithm since it works only on the feature subset of the model [42]. Since the data handled here were healthcare data, which are often

inconsistent, building a predictive model for inconsistent data would have been tedious. However, the random forest algorithm also performs well with conflicting information. Random forest consists of several decision trees, each receiving different data samples. The classification result was obtained by applying majority voting on the predictions from ensemble decision trees. To avoid overfitting in the random forest classification model, the optimal hyper parameter values were chosen through a grid search mechanism. In the grid search algorithm, each hyperparameter is assigned a set of values (grid). Each combination of hyperparameters from the grid is applied to identify the best score for this dataset. Thus, the combination of hyperparameter values obtained with the best score is considered optimal for that hyperparameter. In this current work, hyperparameters included n_estimators, criterion, max_depth, min_sample_leaf, and max_features, resulting in optimal values of 100, entropy, 15, 8, 8, and auto, respectively.

As a result of the grid search, entropy was identified as the suitable criterion to attain high accuracy of the user dataset (entropy equation given in [43].

$$\text{Entropy} = \sum_{i=1}^{C} -p_i \times \log_2(p_i). \tag{5}$$

The well-trained classification model categorized each user into corresponding contextual user groups. When new users arrived at a particular contextual user group, they were correctly classified. These classified data were directly used for the online part. There was another branch in the offline part responsible for spline fitting of user data, where we spline-fitted each data point and the cumulative results for second-level clustering.

### 3.3.4. Spline Fitting of Data

In this process, we took the daily step count data on an hourly basis. For each user, the total step count data were spline-fitted. This process was executed for all users with their historical step count data. Thus, over 1 day, there were 24 data points for each user corresponding to 24 h. This process was used to calculate the parameters in a spline polynomial model. The spline coefficients helped in identifying the optimal field [44].

Firstly, the contextual user groups were given one by one. For each user group, the user data were split on a daily basis. For example, the cumulative step count of Monday's data for one user was taken and spline-fitted. In the next step, all the spline-fitting results for a given user group were combined daily to obtain the graph's spline parameters (Figure 3). This process was applied to all contextual user groups defined by the first-level clustering.

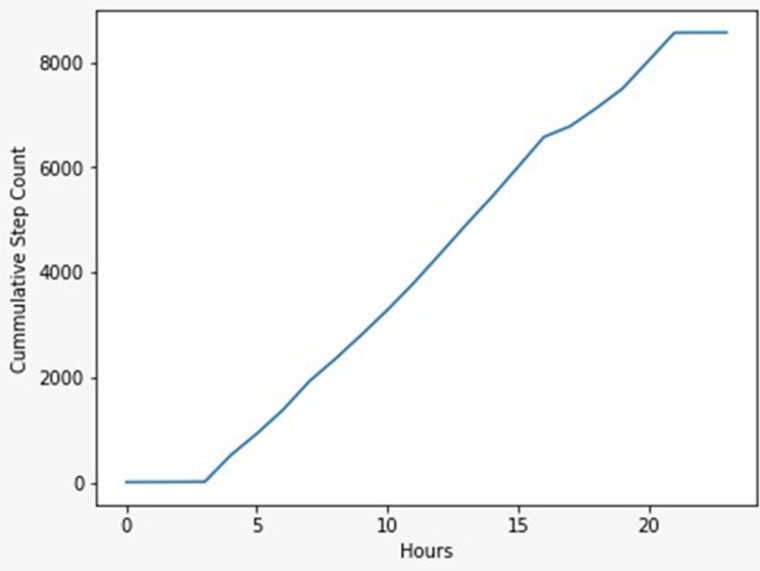

**Figure 3.** Spline-fitted data representation for second-level clustering.

This spline-fitted data were used as input to process the second-level clustering. Spline fitting reduced the noise in the step count trajectories and smoothened the data curve. The 24 data points representing each hour of the day were obtained from the data and joined to form a step count trajectory, featuring both sedentary thresholds and spikes. The passive threshold determined whether the user was inactive for that hour. In order to analyze the user's activity pattern, we used the spline-fitted data to perform the second level of clustering. Then, the statistical measure of the cumulative step count data was extracted from the spline graph.

### 3.3.5. Second-Level Clustering

The second level of clustering took spline-fitted data as the input. This clustering step was performed to divide the users of each contextual user group on the basis of their daily activity pattern. Here, we used the hierarchical agglomerative clustering methodology to cluster each contextual user group on the basis of the time for which the user performed the physical activity. The daily activity pattern of individuals were defined as follows:

1. Morning intense,
2. Afternoon intense,
3. Evening intense,
4. Uniform intense.

The activity pattern for individuals could vary for each day. For instance, a user could have a busy schedule in the morning every Tuesday, whereby their activity pattern could be afternoon or evening intense. Hence, the activity pattern of each user was classified and clustered for each day of the week. This could be used in giving recommendations for each user daily. As previously mentioned, the hierarchical agglomerative clustering methodology was used for this second-level clustering. The silhouette index determined the number of clusters that would be suitable for the process. The silhouette index gave a score for the number of clusters that could be formed, ranging from 1 to $n$.

The use of all four activity patterns in each contextual user group was not mandatory. Hence, the silhouette index score was applied to choose the cluster count to split the contextual user groups according to their activity pattern. This level of clustered data was used to recommend the appropriate activity plan based on the activity pattern. Subsequently, online processing was applied to these second-level clustered data.

### 3.4. Online Computations

An Android application was designed to obtain the necessary data of a new user, classify the user into a specific contextual user group, and spline-fit the user data for second-level clustering. Finally, to provide the recommendations for their activity pattern, recommendations were provided using the linear quadratic regulator (LQR) planner or correct predictions.

Collecting the data from the user: A new user using this application gives their gender, age, height, weight, and BMI as input. Using these data and the step count target, the user is placed into a specific contextual user group. Then, their data are spline-fitted to determine their activity pattern. As previously mentioned, hierarchical aggregation clustering is used for first- and second-level clustering.

Recommendation: All these data are collected using the Android application and given to the backend for further computation. This process is for a new user. In contrast, if an old user is using the application to find their recommended activity graph, the activity graph is generated from their activity pattern. If the user is predicted to fail their daily target, a new graph is generated using the LQR planner.

### 3.4.1. Predictive Model Using Long Short-Term Memory

LSTM has recently become popular across various applications [45]; hence, we applied LSTM here for prediction. LSTM is based on self-looping memory cells and a tri-gated network of input, forget, and output gates [46]. LSTM has a memory cell built in the

memory block to store current and previous timestamps. The input, forget, and output gates can be designed to introduce, discard, and update memory cells. A sigmoid function controls the input gate, which turns the gate on or off according to the current input state and previous output state [47]. As long as the input gate has a value of zero or near zero, updating the memory cell state will not affect the cell. This allows for a more profound network architecture with long short-term memory cells in each layer but with many hidden cells. The extensive computation of a single LSTM cell is governed by Equations (6)–(11).

$$\text{ip\_g}^t = \sigma\left(b_i^{ip_g} + w_{i,j}^{ip_g}\left[h_j^{(t-1)}, \; x_i^{(t)}\right]\right), \tag{6}$$

$$\text{ct}^{(t)} = \tanh\left(b_i^c + w_{i,j}^c\left[h_j^{(t-1)}, \; x_i^{(t)}\right]\right), \tag{7}$$

$$\text{fr\_g}^t = \sigma\left(b_i^{fr\_g} + w_{i,j}^{fr\_g}\left[h_j^{(t-1)}, x_i^{(t)}\right]\right), \tag{8}$$

$$\text{c}^t = fr\_g_i^{(t)} \times c_i^{(t-1)} + ip\_g_i^{(t)} \times \tilde{c}_i^{(t)}, \tag{9}$$

$$\text{ot\_g}^t = \sigma\left(b_i^{ot\_g} + w_{i,j}^{ot\_g}\left[h_j^{(t-1)}, \; x_i^{(t)}\right]\right), \tag{10}$$

$$\text{h}^t = \tanh\left(c_i^{(t)} \times ot\_g_i^{(t)}\right), \tag{11}$$

where $b^{ig,c}$ and $w^{ig,c}$ are the weight and bias of the input gate and intermediate cell, respectively. The new cell state of the memory unit is updated by integrating the preceding cell state $c_i^{(t-1)}$ (Equations (17)–(22)) and intermediate cell state $\tilde{c}_i^{(t)}$ with the impact of the forget gate $fg_i^{(t)}$ and input gate $ig_i^{(t)}$. In Equation (19), $b^{fg}$ and $w^{fg}$ are the weight and bias of the forget gate, respectively, where the output of the forget gate $fg_i^{(t)} \in (0,1)$ is obtained via the logistic sigmoid function σ, where '0' denotes forget and '1' denotes retain. The output gate is computed using $b^{og}$ and $w^{og}$, which are the weight and bias of the output gate, respectively, whereas the nonlinearity of the LSTM network is captured using activation functions σ and tanh.

In order to explore the periodic features of the resource usage time series data, the temporal output information was associated with a deep Bi-LSTM network for precise prediction in this research. The bidirectional LSTM network operates on the input sequence by stacking two unidirectional LSTMs for forward and backward passes at the same time. As a result, each bidirectional LSTM contains twice the number of memory cells, making it possible to learn from the past and the future [48]. This ability to learn from the past and the future is layered to create the Bi-LSTM architecture, which can predict short-term VM resource usage. For Bi-LSTM, memory cells dedicated to forward and backward passes store information using past and future values, respectively. Each hidden state $\overleftarrow{\text{hidden}}_t$, $\overrightarrow{\text{hidden}}_t$ (Equation (12)) is concatenated to form one active state [49].

$$\text{hidden}^t = \sigma\left(\overleftarrow{\text{hidden}}_t, \; \overrightarrow{\text{hidden}}_t\right). \tag{12}$$

Accordingly, in the forward LSTM unit using the bidirectional approach, the memory cell calculates the first output cell (h1) and calculates the new state (h2). At any time step t, the forward and backward passes of the Bi-LSTM perform the classical long short-term unit operations. In this way, the additional information can be gathered, and a boost in prediction accuracy is achieved by combining the hidden states of both directions as one output. In this fashion, additional features can be captured that boost the prediction accuracy.

In Equation (12), σ denotes the combining function, i.e., addition, multiplication, concatenation, or average. The mathematical equations governing the proposed model's

stacked layer architecture are expressed in Equations (13)–(17). The representation for the LSTM network function is replaced with $\mathbb{LSTM}(.)$.

$$\overset{\leftarrow}{OP}_L^{(t)} = \mathbb{LSTM}\left( \overset{\leftarrow}{fg}_L^{(t)}, \overset{\leftarrow}{ig}_L^{(t)}, \overset{\leftarrow}{og}_L^{(t)}, \overset{\leftarrow}{h}_L^{(t-1)}, IP^{(t)} \right), \tag{13}$$

$$\overset{\rightarrow}{OP}_L^{(t)} = \mathbb{LSTM}\left( \overset{\rightarrow}{fg}_L^{(t)}, \overset{\rightarrow}{ig}_L^{(t)}, \overset{\rightarrow}{og}_L^{(t)}, \overset{\rightarrow}{h}_L^{(t-1)}, IP^{(t)} \right), \tag{14}$$

$$\overset{\leftarrow}{OP}_L^{(t+1)} = \mathbb{LSTM}\left( \overset{\leftarrow}{fg}_L^{(t+1)}, \overset{\leftarrow}{ig}_L^{(t+1)}, \overset{\leftarrow}{og}_L^{(t+1)}, \overset{\leftarrow}{hidden}_L^{(t)}, OP_L^{(t)} \right), \tag{15}$$

$$\overset{\rightarrow}{OP}_L^{(t+1)} = \mathbb{LSTM}\left( \overset{\rightarrow}{fg}_L^{(t+1)}, \overset{\rightarrow}{ig}_L^{(t+1)}, \overset{\rightarrow}{og}_L^{(t+1)}, \overset{\rightarrow}{hidden}_L^{(t)}, OP_L^{(t)} \right), \tag{16}$$

$$FCL^{t+1} = \left( W_{\overset{\leftarrow}{h}} * \overset{\leftarrow}{OP}_L^{(t+1)} + W_{\overset{\rightarrow}{h}} * \overset{\rightarrow}{OP}_L^{(t+1)} \right) + b_{FCL}, \tag{17}$$

where *fg*, *ig*, and *og* denote the output of the tri-gate operation of a single cell in the LSTM network. $\overset{\leftarrow}{OP}_L^{(t)}$ in Equation (3) and $\overset{\rightarrow}{OP}_L^{(t)}$ in Equation (4) are outputs from the forward and backward directions of the first Bi-LSTM layer of the deep network, respectively, whereby $OP^{(t)}$ in Equation (3) is the input of the deep Bi-LSTM, whereas $\overset{\leftarrow}{OP}_{L+1}^{(t+1)}$ in Equation (4) and $\overset{\rightarrow}{OP}_{L+1}^{(t+1)}$ in Equation (5) are bidirectional outputs from the second Bi-LSTM of the deep network. The last fully connected layer is demonstrated as $FCL^{t+1}$ in Equation (7), which is treated as the final output from the Conv deep Bi-LSTM for cloud resource demand prediction.

To explore the hyperparameter tuning of LSTM, the grid search mechanism was chosen in this work. The final optimal values of hyperparameters were as follows: 50 hidden neurons, three hidden layers, learning rate of 0.001, batch size of 1200, and 75 epochs. The dropout layer ensured the establishment of model generalization and avoidance of overfitting. Lastly, the mapping of input resource usage time series features to the output future demand forecast was accommodated by a fully connected and regression layer, collectively known as a dense layer. LSTM was used to predict whether the user could achieve their daily target or not. If a person achieves the daily target, this model does not call the LQR. If the person fails to achieve the daily target, then the LQR planner is utilized to recommend a new activity graph [50]. Thus, this complete model can help a person to achieve their daily step count target and adhere to their physical activity regimen. The linear quadratic regulator recommends a new plan if the predictive model returns a value of 0. LQR, thus, adjusts the step count goals for the remainder of the day according to the cumulative step count up until that hour and the step difference.

### 3.4.2. LQR Planner

The linear quadratic regulator is called when the predictive model returns a value of 0, indicating that the user will not achieve their goal. The LQR changes the step count goals for the remainder of the day on the basis of the cumulative step count up until that hour, and the step difference is adjusted accordingly. Otherwise, this method is not called. LQR helps the user to achieve their step count target optimally [35,51]. It works in different states. Firstly, it takes the actual state or actual step count into consideration. Then, if the predictive model result is 0, it determines the desired state. It then works out the state cost

matrix and the input cost matrix. The role of LQR here is to reduce the state error cost in any way (Equation (18)).

$$
\begin{aligned}
&\text{LQR (Actual State x, Desired State xf, Q, R, A, B, dt):}\\
&\text{x\_error = Actual State x} - \text{Desired State xf; Initialize N = \#\#,}
\end{aligned}
\tag{18}
$$

where Q is the cost matrix, R is the input cost matrix, and A and B are the two states. This method helps in recommending the perfect activity graph for the person. This total system was developed as an Android application to make it accessible to the users [7]. All these functionalities were incorporated into the application, giving users personalized recommendations.

## 4. Experimental Results and Discussion

### 4.1. Data Preprocessing

The offline part of the project included data preprocessing, two-level clustering, and a spline-fitting model. In the preprocessing data stage, we gathered and processed the data in the desired formats for the machine learning models. This was followed by two levels of clustering and spline fitting of the dataset to help the models in clustering. We describe the results of each step mentioned above in detail.

We start with the dataset description. As mentioned earlier, the dataset used for this project could be split into two different types of data: physiological features and physical features.

The physical feature dataset consisted of the step counts and calories burnt for 533 users over 42 days. Among the 533 users, 290 were male, while 243 were female. The step count dataset contained the hourly step count of each user for 42 days indexed on a daily basis. The calories burnt dataset was analogous to the step count data. Figure 4 shows the sample step count data, while Figure 5 shows the corresponding calorie data of the user over 5 days. Figure 6 depicts the sample dataset for calories burnt.

The physiological feature dataset consisted of the users' gender, age, height, weight, and BMI. Data from both these datasets were used to prepare a contextual feature dataset containing the physiological features and the data computed from the physical features of the users. A few data cleaning techniques were then applied to the dataset. Rows with more than 30% missing data were removed, while the remaining null values were interpolated linearly. Outliers were also removed such that the data did not affect the machine learning models during the training phase. Figure 5 shows a sample of the final contextual data used for first-level clustering.

| ID | Gender | Age | Height | Weight | BMI | Workout Par Rate | Average Workout Time | STD Workout Time | Average Calories Burnt | STD Calories Burnt | Number of Sessions |
|---|---|---|---|---|---|---|---|---|---|---|---|
| 1001 | Female | 23 | 1.60 | 52 | 20.312500 | 0.700000 | 1.476190 | 0.578565 | 304.000000 | 131.362820 | 21 |
| 1002 | Male | 19 | 1.80 | 80 | 24.691358 | 0.800000 | 1.333333 | 0.384900 | 276.875000 | 94.906698 | 24 |
| 1003 | Male | 20 | 1.83 | 66 | 19.707964 | 0.866667 | 1.423077 | 0.504700 | 302.538462 | 121.595039 | 26 |
| 1004 | Male | 25 | 1.78 | 78 | 24.618104 | 0.733333 | 1.454545 | 0.548202 | 310.727273 | 131.112156 | 22 |
| 1005 | Female | 19 | 1.64 | 53 | 19.705532 | 0.766667 | 1.391304 | 0.461557 | 288.608696 | 112.539132 | 23 |

**Figure 4.** Sample of step count dataset.

| Day | 00-01 hr | 01-02 hr | 02-03 hr | 03-04 hr | 04-05 hr | 05-06 hr | 06-07 hr | 07-08 hr | 08-09 hr | 09-10 hr | ... | 15-16 hr | 16-17 hr | 17-18 hr | 18-19 hr | 19-20 hr | 20-21 hr | 21-22 hr | 22-23 hr | 23-24 hr | Total Steps |
|---|---|---|---|---|---|---|---|---|---|---|---|---|---|---|---|---|---|---|---|---|---|
| Tuesday | 5 | 2 | 3 | 2 | 18 | 5280 | 3168 | 1056 | 528 | 438 | ... | 145 | 16 | 187 | 42 | 118 | 128 | 58 | 0 | 4 | 11682 |
| Wednesday | 3 | 2 | 0 | 0 | 113 | 6374 | 3824 | 1274 | 637 | 283 | ... | 51 | 187 | 18 | 83 | 101 | 26 | 45 | 3 | 3 | 13487 |
| Thursday | 5 | 1 | 0 | 4 | 46 | 6735 | 4041 | 1347 | 673 | 55 | ... | 183 | 56 | 123 | 130 | 14 | 153 | 169 | 0 | 1 | 14249 |
| Friday | 5 | 4 | 4 | 0 | 29 | 7121 | 4272 | 1424 | 712 | 656 | ... | 83 | 125 | 52 | 9 | 166 | 109 | 3 | 5 | 4 | 15238 |
| Saturday | 4 | 4 | 2 | 3 | 2 | 147 | 133 | 70 | 110 | 55 | ... | 117 | 38 | 227 | 226 | 206 | 246 | 251 | 334 | 2 | 2866 |

**Figure 5.** Sample of contextual features dataset (for Total Steps).

| | 00-01 hr | 01-02 hr | 02-03 hr | 03-04 hr | 04-05 hr | 05-06 hr | 06-07 hr | 07-08 hr | 08-09 hr | 09-10 hr | ... | 15-16 hr | 16-17 hr | 17-18 hr | 18-19 hr | 19-20 hr | 20-21 hr | 21-22 hr | 22-23 hr | 23-24 hr | Total Calories |
|---|---|---|---|---|---|---|---|---|---|---|---|---|---|---|---|---|---|---|---|---|---|
| **Day** | | | | | | | | | | | | | | | | | | | | | |
| Tuesday | 0 | 0 | 0 | 0 | 0 | 211 | 126 | 42 | 21 | 17 | ... | 5 | 0 | 7 | 1 | 4 | 5 | 2 | 0 | 0 | 467 |
| Wednesday | 0 | 0 | 0 | 0 | 4 | 254 | 152 | 50 | 25 | 11 | ... | 2 | 7 | 0 | 3 | 4 | 1 | 1 | 0 | 0 | 539 |
| Thursday | 0 | 0 | 0 | 0 | 1 | 269 | 161 | 53 | 26 | 2 | ... | 7 | 2 | 4 | 5 | 0 | 6 | 6 | 0 | 0 | 569 |
| Friday | 0 | 0 | 0 | 0 | 1 | 284 | 170 | 56 | 28 | 26 | ... | 3 | 5 | 2 | 0 | 6 | 4 | 0 | 0 | 0 | 609 |
| Saturday | 0 | 0 | 0 | 0 | 0 | 5 | 5 | 2 | 4 | 2 | ... | 4 | 1 | 9 | 9 | 8 | 9 | 10 | 13 | 0 | 114 |

**Figure 6.** Sample of contextual features dataset (for Total Calories).

*4.2. Offline Computation Results*

**First-level clustering:** We performed clustering on the contextual feature dataset after the preprocessing step. We used the hierarchical agglomerative clustering algorithm to cluster the preprocessed data. The silhouette index was used to evaluate the objects' similarity to their cluster. Figure 7 shows the silhouette scores of different numbers of clusters. It can be observed that, when the dataset was clustered into three contextual groups, the silhouette index had the greatest value (>0.8). On this basis, we clustered the data into three contextual groups.

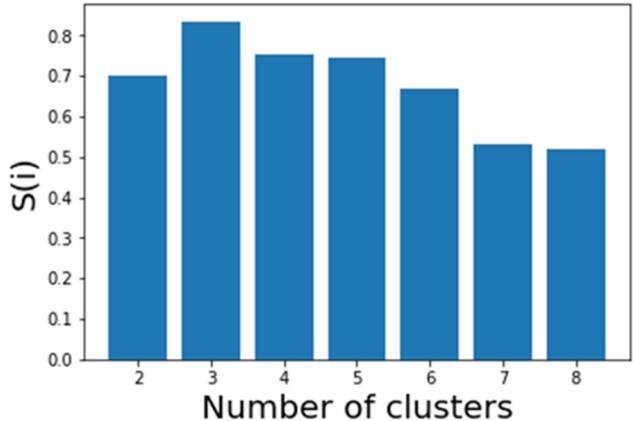

**Figure 7.** Silhouette index for different numbers of clusters.

**User classification results:** After the users were clustered into different contextual groups, we determined the group into which a new user would fall. We built a classification model to classify new users into a contextual group on the basis of their features received through the Android application. Since the user entered their data into the system, they could be classified in the online part. In the offline part, we trained the model using the data from the users in the dataset.

For the classification, we used the random forest ensemble algorithm. Since users could not provide all details, the random forest processed the information in two ways: data with partial features and data with complete features. Users with a history of daily activity records were classified by complete features. It can observed that, since we used a lower number of attributes for partial features, the accuracy was comparatively low. We found that classifying a user on the basis of their complete features resulted in an accuracy of 0.97 and a recall of 0.96, whereas classifying them on the basis of their partial features resulted in an accuracy of 0.82. We stored this model in the mobile application to classify new users (Table 4).

**Table 4.** Number of activity pattern groups clustered.

| Contextual Group | Group 1 | Group 2 | Group 3 |
|---|---|---|---|
| Monday | 2 | 5 | 3 |
| Tuesday | 2 | 4 | 3 |
| Wednesday | 4 | 7 | 2 |
| Thursday | 4 | 4 | 2 |
| Friday | 4 | 5 | 2 |
| Saturday | 2 | 3 | 4 |
| Sunday | 2 | 2 | 3 |

**Spline fitting:** Each user's daily average step count was spline-fitted. Features were engineered from the fitted spline for a second-level clustering model.

**Second-level clustering:** We set the spike threshold to 150, i.e., anything above 150 calories burnt per hour was considered a spike, whereas anything below 15 calories burnt was considered sedentary. According to the daily activity patterns of the users, we clustered them into the contextual groups using the spike sedentary features and the engineered features from the spline fitting.

It is evident from Figure 8 that a person's daily activity pattern was not necessarily uniform across all days of the week. For example, a user could be morning intensive on Monday but evening intensive on Tuesday. We performed second-level clustering for the users according to the day of the week.

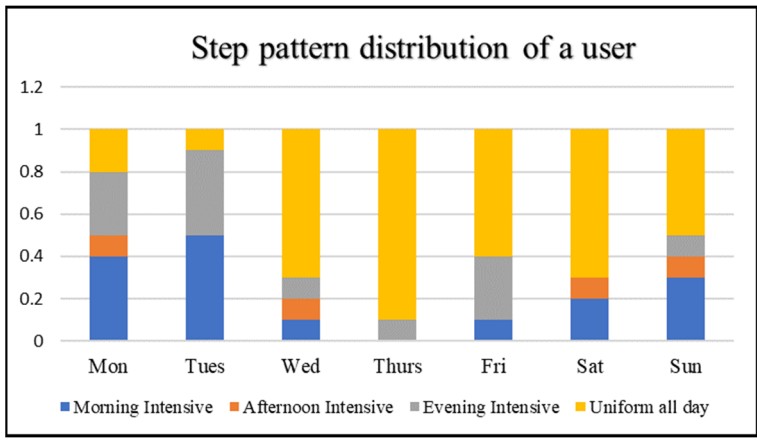

**Figure 8.** Daily step pattern of a user across all days of the week.

### 4.3. Online Mode Computation Results

The online part of the project included the process of classifying new users on the basis of their features, a predictive model that predicted if the user would achieve their daily target, and an LQR planner to modify the recommended activity pattern if it was predicted that the user would not achieve their target.

**User classification:** Using the random forest model, the users were classified according to their contextual features in the previous section. This was performed for all users who registered through the mobile application. After the user entered sufficient details, they were classified using the already trained RF model.

**Daily activity recommendation:** The user was classified into any of the three contextual groups according to their history of daily activity patterns. If the user was new, a daily activity pattern most frequently observed for each day of the week was recommended.

**Deep learning prediction results:** After users receive their recommended activity patterns, it is very important to monitor if their adherence. We considered different

algorithms such as random forest (RF), Support vector regression (SVR), recurrent neural network (RNN), and long short-term memory (LSTM) for prediction. On comparing all the models, we found that LSTM outperformed other algorithms.

- Support vector regression: SVR works similarly to SVM classification but is applied to predict real values instead of classes.
- Recurrent neural network: In this neural network, the output from one step is fed into the next step. In this context, the output of the same day from the previous week was used as the input for the next week.
- Random forest: This ensemble model is similar to the classification model.
- Long short-term memory: This is a deep learning variation of RNN that utilizes the output from the previous step as input.

We experimented with all four abovementioned methods and compared the results using the following metrics: accuracy of the model, mean absolute error (MAE; magnitude of the difference between the predicted and true values of the observation), root-mean-square error (RMSE; squared error used as the loss function), and *R*-squared ($R^2$; the amount of variance in the predictions).

Table 5 shows the comparison of the machine learning models with their result metrics. We found that LSTM outperformed all the other models for our dataset. We trained the LSTM model and stored the model in the Android application. At the end of every hour, the model checks if the user will achieve their daily goal. Since this step is crucial for recommendation, we experimented with different models, eventually selecting LSTM.

**Table 5.** Comparison of different ML models.

| Models | Accuracy | RMSE | $R^2$ | MAE |
|--------|----------|------|-------|-----|
| SVR | 86.18 | 1.23 | 0.58 | 1.1 |
| RNN | 92.33 | 0.651 | −7.35 | 0.785 |
| RF | 94.47 | 0.521 | 0.75 | 0.66 |
| LSTM | 97.8 | 0.38 | 0.96 | 0.594 |

The linear quadratic regulator is called when the predictive model returns a 0 (indicating that the user will not achieve their daily goal). LQR then changes the step count goals for the remainder of the day according to the cumulative step count up until that hour and the step difference.

### 4.4. Android Application

We developed an android application through which we obtained the required details from the user and then used the algorithms from the previous section to recommend a daily activity pattern. This application uses java as the backend, accompanied by the appropriate libraries to achieve a neat user interface. We also use Firebase and Firestore for Google authentication and storage of the user data.

Figure 9 shows the login page of the Android application, revealing the email ID and password text fields. Users can either log in using their account or sign up via Google. The second image on the right shows the Google sign-up page where the list of emails is blurred for privacy reasons. When a new user uses this app, they can register their account using the "Register Here" button.

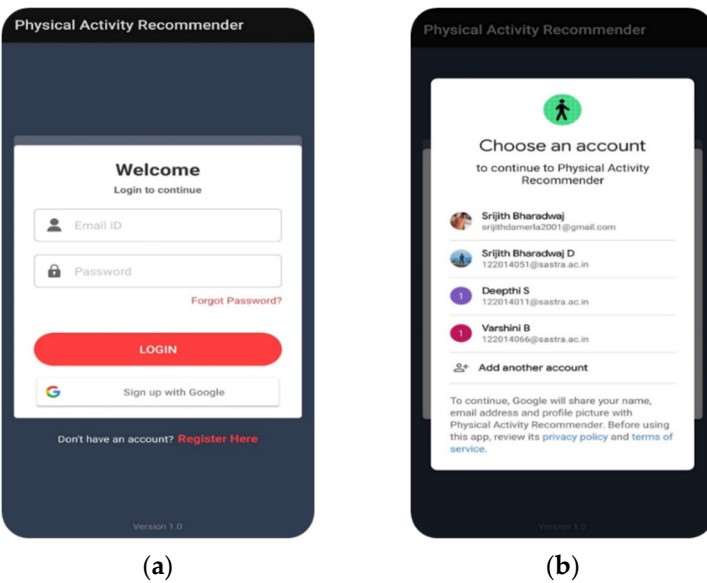

**Figure 9.** Login page of the android application: (**a**) welcome page with user credentials; (**b**) Google sign-up option for login using user's Gmail account.

Figure 10 shows the page of the Android application where a new user can enter their details and register for the recommendation system application. When a user clicks the "Register" button, details from the text fields are stored in the Firestore database. These details are fetched when the user logs into the app. Figure 11 shows a sample view of the Firebase console after successfully registering for the application. After successful registration, the user is directed to the user detail collection page where they can enter the required details such as gender, height, weight, and daily step goal. Users are only directed to this page if their historical data are not stored in our database or in the Google fit application. Once the user submits their data, they are classified into one of the three clusters. The pretrained model is deployed in the application to classify new users. After successfully classifying the users, the most frequent activity pattern from the contextual group is used to calculate the hourly step goal on the basis of the user's daily step goal. Figure 12 shows the recommended activity pattern of a sample user, tested using our application.

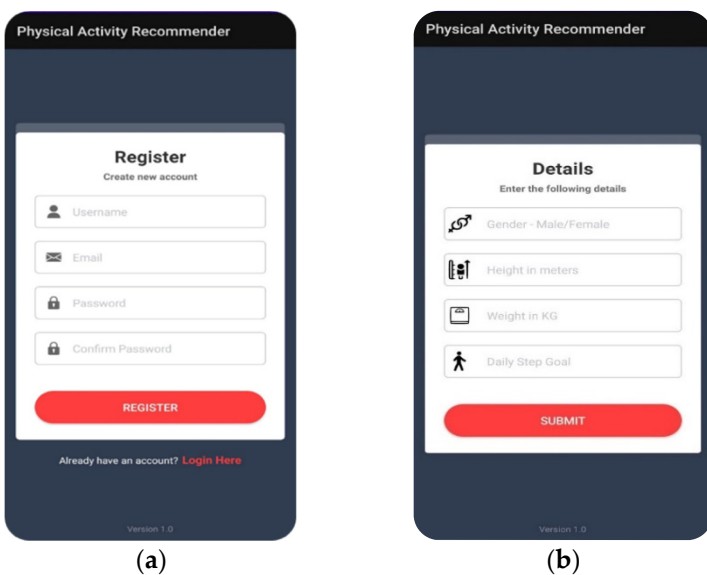

**Figure 10.** Recommended activity pattern and step count progress tracker: (**a**) register page of the android application; (**b**) user detail collection page.

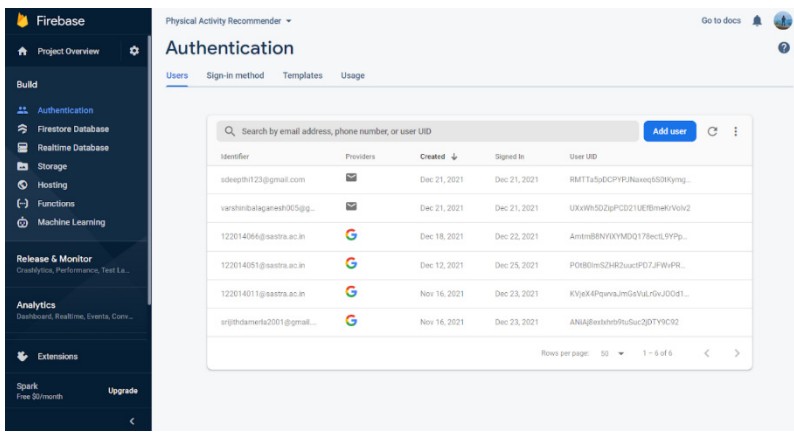

**Figure 11.** Firebase console for Google authentication.

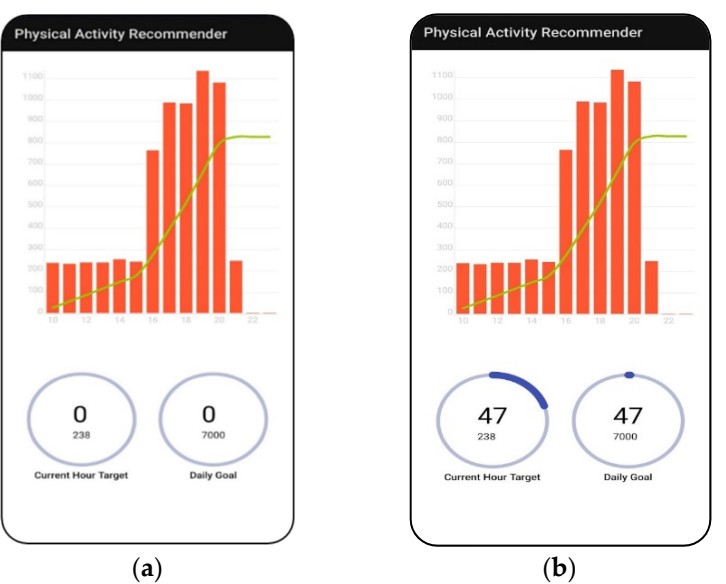

(**a**)                    (**b**)

**Figure 12.** (**a**) Recommended activity pattern; (**b**) step count progress tracker.

The left circular progress bar shows the progress of the user's current hourly step target, while the right circular progress bar shows the progress toward their daily step goal. The graph and the spline curve are dynamically produced according to the hour of the day. The graph shows the step count target for the remaining hours of the day, while the curve shows the average step target for every hour. The user's hourly step count is stored in the Firestore and used in the forthcoming weeks for a precise recommendation to the user based on their activity pattern in the previous week.

Figure 12 shows the step count progress tracker, which uses a pedometer sensor to capture the steps taken by the user. This application runs in the background to minimize user intervention. At the end of every hour of the day, details of the step count, step difference, and cumulative steps, along with other features, are provided to the predictive model.

This is a crucial step of the recommendation system since the step target pattern depends on the result of the prediction. If the output of the predictive model is 0, the LQR planner is used to adjust the graph and the target for the remainder of the day. This step is repeated at the end of every hour. At the end of every day, a new graph is recommended according to the day of the week, and all details from the previous day are stored in the Firestore.

## 5. Discussion

This research focused on developing a deep learning-based physical activity support system to generate recommendations through a mobile fitness application. The major challenge lies in tracking each user's step count pattern dynamically according to their daily life activities. Thus, the proposed method constructed two-level clustering algorithms to classify the users according to contextual group and daily activity. For the contextual group, the random forest deep classification model classifies the user group using both partial and complete features. A critical element of the application is in engaging both adherent and nonadherent users via a physical activity tracking system. In the case of adherent users, a widely used deep learning prediction model forecasts their successive hourly step count for precise encouragement. For nonadherent users, the LQR planner adjusts the daily target of the user through the physical activity tracker, while suggesting recommended physical activities. This was made possible by developing a new Android application incorporating the proposed physical activity support system.

## 6. Conclusions and Future Plans

An adaptive physical activity predictor and advisor can help users to lead a healthy life by adhering to a physical activity regimen. With the help of physical activity data gathered from a huge fitness tracker population, clusters of user groups were formed. Our system discovers patterns from the everyday activity of each user in these groups and uses those patterns to set future activity plans for users. After tracking user's activities, notifications based on their probability of achieving their goals are sent to them. The LQR planner generates an alternate activity plan when the probability of meeting these goals is low.

In first-level clustering, agglomerative hierarchical clustering groups users into contextual groups according to their physiological and calculated physical features. Agglomerative clustering is easy to implement, and it works on the basis of the dissimilarities between the objects to be grouped. The silhouette score was greatest for three clusters; hence, three contextual groups were chosen for the dataset. For the user group classification model, we used the random forest ensemble algorithm. For the partial features classification model, we attained an accuracy of 0.82, whereas, for the complete features classification model, we attained an accuracy of 0.97. The model was spline-fitted and used for second-level clustering, again using agglomerative hierarchical clustering, this time to group users according to daily activity. These clusters were used for the online part of the project. For the predictive model, the deep learning approach LSTM had the highest accuracy of 97.8%, followed by other algorithms such as SVM, RNN, and random forest with accuracies of 86.18%, 92.33%, and 94.47%, respectively. When the LQR model returned a value of 0, the users are recommended changes in their activity plan. The whole process was developed into an Android application to make it more user-friendly. Incorporating a drift detection and an automated model into the framework represent potential future improvements. Issues related to data security should also be addressed.

**Author Contributions:** Conceptualization, S.V., V.V., D.S., V.B., S.B.D., B.S. and L.R.; methodology, S.V., V.V., D.S., V.B., S.B.D., B.S. and L.R.; investigation, S.V., V.V., D.S., V.B., S.B.D., B.S. and L.R.; writing—original draft preparation, S.V., V.V., D.S., V.B., S.B.D., B.S. and L.R.; writing—review and editing, S.V., V.V., D.S., V.B., S.B.D., B.S. and L.R. All authors have read and agreed to the published version of the manuscript.

**Funding:** This research received no external funding.

**Institutional Review Board Statement:** Not applicable.

**Informed Consent Statement:** Not applicable.

**Data Availability Statement:** The data that support the findings of this study are available on request from the corresponding author.

**Acknowledgments:** The authors gratefully acknowledge the Science and Engineering Research Board (SERB), Department of Science and Technology, India for the financial support through the Mathematical Research Impact Centric Support (MATRICS) scheme (MTR/2019/000542). The authors also acknowledge SASTRA Deemed University, Thanjavur, for extending the infrastructural support used to carry out this research.

**Conflicts of Interest:** The authors declare no conflict of interest.

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
