# Peer review of "Dynamic Physical Activity Recommendation Delivered through a Mobile Fitness App: A Deep Learning Approach"

_axioms, doi:10.3390/axioms11070346_

Round 1

Reviewer 1 Report

This paper proposes a machine learning approach for physical activity. Despite the interesting approach, the work lacks evidencing novelty, there is a misplacement of information among the topics, and the description of some procedures should be significantly improved.

My main concerns are:

·         The writing does not follow a technical pattern and I strongly recommend the authors to use a grammar English service.

·         The novelty of this study is not evident. The introduction should contain very explicitly the objectives of this study.

·         There is a misplacement of information in this manuscript. Some information in the introduction should be placed in the materials and method section, such as “Due to improvements in technology such as digital watches and mobile phones that can more precisely forecast data, collecting real-time statistics with high precision has become simple. The accelerometer data for 533 individuals, 290 of whom were men and 243 of whom were females, was utilized in this investigation.”

·         The data description is poorly described. It would be critical to show the range of values for each feature. Besides, it is not described how the data was partitioned for training, validation, and testing.

·         The entire methodological section should include more details.

·         The explanation of the ML and DL approaches is insufficient. There is no explanation regarding the hyperparameters. For example, what was the learning rate of the LSTM model? Other models should also be compared, for example, the XGBoost.

·         “For the classification, we have used the Random Forest ensemble algorithm. Since users cannot provide all the details, we have trained two classification models – one with partial features; the other with complete features.” The XGBoost model can learn with missing data, it might be a good choice. Moreover, why didn’t the authors use other methods for filling the empty data?

·         There is no discussion. The authors should use a discussion section to clearly state the novelty of this study. How does it differ from other applications? This should be highlighted.

·         “We set the spike threshold at 150 i.e., anything above 150 calories burn per hour is considered a spike, and the sedentary threshold at 15 o anything below 15 is considered sedentary” The authors made that decision based on what? Are there studies that determine those numbers? It is not clear to the readers.

·         All figures and tables must be self-explanatory. Thus, the authors must inform what the acronyms and abbreviations mean.

·         For the review process, it would be very helpful if the authors put line numbers.

·         The MDPI papers have templates. The authors should use the template of this journal.

Some minor notes

Abstract

The abstract is probably the first thing the readers will read, and it should contain some critical elements from the research. The text does not make it clear what was the research gap, objectives, and results.

Some adjustments could have been made for easier reading, as shown below:

Change “It is essential for good health and mental well-being to adhere to a fixed physical activity regimen.” To “Adhering to a fixed physical activity regimen is essential for good health and mental well-being”

“In this research work” - Either use “research” or “work”.

Change “Based on the user's historical activity habits and current activity objective, the likelihood of sticking to the plan is customized.” To “The likelihood of sticking to the plan is customized based on the user's historical activity habits and current activity objective”

Literature Review

“Physical activity will improve our health exponentially and reduce the risk of various diseases like type 2 diabetes, cancer, and cardiovascular disease” the authors should cite this.

The literature review should not contain information about the current study such as “Hence, this project helps users follow a daily exercise routine to maintain their consistency by helping users to achieve their daily targets even in their busy lifestyles. We inspired from the previous research work from [9] for predicting physical activity adherence based on their step count. Therefore, we proposed a model based on deep learning approach to predict and provide personalized health advisor”

Data pre-processing

“The pre-processed data is free of all the missing values, data inconsistency, and outliers” Why did the authors train two models if you stated that there was no missing values?

Offline Computations

“Impurities accounted for around 30% of the total and were eliminated.” The authors must be more specific.

User classification: Random Forest

Why was this model chosen? What parameters were used? How the data was distributed? A critical part of machine learning models is to optimize. The authors do not even mention validation and testing data in the entire manuscript. This is a critical part.

Deep learning prediction results

The model explanation here is very shallow and insufficient. Besides, the description of models should belong to the methods section, not here.

The methods section should also include a subsection named evaluation metrics, where the authors should make a better description of the metrics.

Why didn’t the authors test different models for the first part as well?

Author Response

Dear Editor-in-Chief, Editors and Reviewers,

Thank you for your useful comments and suggestions to revise our manuscript. We have carefully revised the manuscript accordingly. The detailed revisions/corrections that have been undertaken and our detailed point by point response to all the comments given by the reviewers are listed below for your kind review.

Comments and Suggestions for Authors

This paper proposes a machine learning approach for physical activity. Despite the interesting approach, the work lacks evidencing novelty, there is a misplacement of information among the topics, and the description of some procedures should be significantly improved.

My main concerns are:

Reviewer#1, Concern # 1: The writing does not follow a technical pattern and I strongly recommend the authors to use a grammar English service.

Author response:  Thank you for your comment. As per the reviewer’s valuable suggestions and instruction, the organization of the paper is modified, as well as the grammar mistakes and typos are double checked and corrected. Throughout the manuscript corrections are made.

Reviewer#1, Concern # 2: The novelty of this study is not evident. The introduction should contain very explicitly the objectives of this study.

Author response:  Thank you for your valuable comment. The author updated the introduction section by updating the objectives and novelty of the study in a clear manner.

Reviewer#1, Concern # 3: There is a misplacement of information in this manuscript. Some information in the introduction should be placed in the materials and method section, such as “Due to improvements in technology such as digital watches and mobile phones that can more precisely forecast data, collecting real-time statistics with high precision has become simple. The accelerometer data for 533 individuals, 290 of whom were men and 243 of whom were females, was utilized in this investigation.”

Author response:  Thank you for your observation. As per the valuable advice of reviewer, the content have been rearranged from introduction to the dataset description section and highlighted.

Reviewer#1, Concern # 4: The data description is poorly described. It would be critical to show the range of values for each feature. Besides, it is not described how the data was partitioned for training, validation, and testing.

Author response:  Thank you for your valuable comment. As per the concern, the sample dataset has been included as table to show the ranges of each physical and physiological features.

Reviewer#1, Concern # 5: The entire methodological section should include more details.

Author response: Thank you for your observation We have carefully revised the paper as per your comments given and all the changes have been highlighted in red throughout the methodology section 3.

Reviewer#1, Concern # 6: The explanation of the ML and DL approaches is insufficient. There is no explanation regarding the hyperparameters. For example, what was the learning rate of the LSTM model? Other models should also be compared, for example, the XGBoost.

Author response:  Thank you for your comment. As per the valuable advice of the reviewer, the author has been updated the random forest and LSTM content with list of hyper parameters and its values in section 3. As the random forest consist of trees, in this current work, random forecast classification algorithm is compared with Decision Tree to justify the outperformance of ensemble of decision tress in random forest. According to this work, random forest is compared with ML methods like SVM and decision tree. In future analysis, we will do comparison with XGBoost algorithm for the same work.

Reviewer#1, Concern # 7: “For the classification, we have used the Random Forest ensemble algorithm. Since users cannot provide all the details, we have trained two classification models – one with partial features; the other with complete features.” The XGBoost model can learn with missing data, it might be a good choice. Moreover, why didn’t the authors use other methods for filling the empty data?

Author response:  Thank you for your observation. The main motive of this work is to predict the user physical activity for successive rate and providing it as android product. On basis of this idea, user details may not be complete in nature when using android device. As the basic personal details about each user cannot be filled, we aim to implement a classification model which should accepts both partial and complete features. It is successful by the implementation of random forest algorithm where the features have been selected based on their importance. Still the work two classification model is removed and updated in the way of projecting like two way of classication using two different features. We have included the mentioned comments in the random forest user classification subheadings.

      Reviewer#1, Concern # 8: There is no discussion. The authors should use a discussion section to clearly state the novelty of this study. How does it differ from other applications? This should be highlighted.

Author response:  Thank you for your input. As per the reviewer’s suggestion, included discussion about the work is included within the conclusion part section 5.

      Reviewer#1, Concern # 9: “We set the spike threshold at 150 i.e., anything above 150 calories burn per hour is considered a spike, and the sedentary threshold at 15 o anything below 15 is considered sedentary” The authors made that decision based on what? Are there studies that determine those numbers? It is not clear to the readers.

Author response:  Thank you for your comment. The authors meant the reference paper for consideration of that numerical value and the changes have been updated in revised manuscript.

      Reviewer#1, Concern # 10: All figures and tables must be self-explanatory. Thus, the authors must inform what the acronyms and abbreviations mean.

Author response:  Thank you for your comment. As per the reviewer’s valuable suggestions, all figures and tables have been updated for easy understanding and the abbreviation are included for all acronyms in first attempt.

      Reviewer#1, Concern # 11: For the review process, it would be very helpful if the authors put line numbers.

Author response:  Thank you for your valuable comment. The author re-edited the whole manuscript and inserted the line numbers.

Reviewer#1, Concern # 12: The MDPI papers have templates. The authors should use the template of this journal.

 Author response:  Thank you for your comment. As per the valuable suggestion of the honorable reviewer, the author have followed the template of the journal and formatted the revised manuscript.

Some minor notes

Abstract

The abstract is probably the first thing the readers will read, and it should contain some critical elements from the research. The text does not make it clear what was the research gap, objectives, and results.

Some adjustments could have been made for easier reading, as shown below:

Change “It is essential for good health and mental well-being to adhere to a fixed physical activity regimen.” To “Adhering to a fixed physical activity regimen is essential for good health and mental well-being”

“In this research work” - Either use “research” or “work”.

Change “Based on the user's historical activity habits and current activity objective, the likelihood of sticking to the plan is customized.” To “The likelihood of sticking to the plan is customized based on the user's historical activity habits and current activity objective”

Author response:  Thank you for your comment. As per the valuable suggestion of the honorable reviewer, the author revised the abstract section for clear understanding. As well as, the mentioned suggestions are replaced in the revised manuscript.

Literature Review

“Physical activity will improve our health exponentially and reduce the risk of various diseases like type 2 diabetes, cancer, and cardiovascular disease” the authors should cite this.

The literature review should not contain information about the current study such as “Hence, this project helps users follow a daily exercise routine to maintain their consistency by helping users to achieve their daily targets even in their busy lifestyles. We inspired from the previous research work from [9] for predicting physical activity adherence based on their step count. Therefore, we proposed a model based on deep learning approach to predict and provide personalized health advisor”

 Author response:  Thank you for your comment. As per the valuable suggestion, the citation have been included for the above sentence and the content related to proposed have been removed from the literature review section.

Data pre-processing

“The pre-processed data is free of all the missing values, data inconsistency, and outliers” Why did the authors train two models if you stated that there was no missing values?

  Author response:  Thank you for your comment. Here the missing values represent with respect to both partial and complete dataset. According to this work, the partial dataset is created because of the user response towards the activity tracker app. The absence of missing values represents the pre-defined partial data are given by user without missing field.

Offline Computations

“Impurities accounted for around 30% of the total and were eliminated.” The authors must be more specific.

Author response:  Thank you for your comment. For this valuable observation, the author have included the corresponding citation at the end of the line.

User classification: Random Forest

Why was this model chosen? What parameters were used? How the data was distributed? A critical part of machine learning models is to optimize. The authors do not even mention validation and testing data in the entire manuscript. This is a critical part.

 Author response:  Thank you for your valuable comment. As per the suggestions, the Random Forest related section is fully re-written with the justifications for the above-mentioned questions. The ratio of training, validation and testing part of dataset is mentioned in the same section.

Deep learning prediction results

The model explanation here is very shallow and insufficient. Besides, the description of models should belong to the methods section, not here.

The methods section should also include a subsection named evaluation metrics, where the authors should make a better description of the metrics.

 Why didn’t the authors test different models for the first part as well?

Author response:  Thank you for your valuable comment. As per the reviews, the LSTM content is updated with the suggestions in the proposed section Since only two algorithms have been explored here, we didn’t take that into separate materials and methods section.

Reviewer 2 Report

The proposal presents a dynamic physical activity recommendation delivered through a mobile fitness app, a deep learning approach. The proposal presents an actual and interesting topic; however, the following aspects were identified:

1) In the introduction it is necessary to describe the acronym CDC, it is recommended that the first time an acronym is used it is described first. The same applies to the acronym LSTM in the "Literature Review" section. In addition, other acronyms that should be described are BMI, BMI.

2) It is noted that a literature review of works related to the proposal is presented; however, the main differences with respect to these works that allow identifying and highlighting the importance and novelty of this proposal are not indicated. In addition, it is suggested that this section be further enriched with a greater number of related works, preferably covering the last 7 years.

3) It is suggested that after presenting a figure there should not immediately be a subtitle or section. This happens when presenting figures 1, 2, 3 and 12, so it is suggested to take care of this detail.

4) In the section "Hierarchical Agglomerative clustering" the formulas for Euclidean distance and Agglomerative clustering are presented, however, the wording does not refer to either formula.

5) Why Random Forest was used and not another classification algorithm?, it is suggested to justify this in the paper.

6) It is noted that in the section "Second level clustering" reference is made to figure 3, however figure 3 is presented in the previous section, so it is suggested that first reference be made to the figure and then the figure is presented. It is suggested to take care of this aspect with all the figures or tables presented in the document.

7) In the Data Pre-processing, Figure 6 is presented, however, the figure is not referenced anywhere in the document.

8) Make sure that the name of the figure 8 is on the same sheet as the figure.

9) It is suggested that in the conclusions or in the discussion part, the main challenges or challenges identified with respect to people who exercise regularly or periodically should be described, but above all how to encourage people who are completely sedentary.

10) It is suggested to review and take care that the format required by the journal is complied with. (https://www.mdpi.com/journal/axioms/instructions )

Author Response

Dear Editor-in-Chief, Editors and Reviewers,

Thank you for your useful comments and suggestions to revise our manuscript. We have carefully revised the manuscript accordingly. The detailed revisions/corrections that have been undertaken and our detailed point by point response to all the comments given by the reviewers are listed below for your kind review.

Comments and Suggestions for Authors

The proposal presents a dynamic physical activity recommendation delivered through a mobile fitness app, a deep learning approach. The proposal presents an actual and interesting topic; however, the following aspects were identified:

1) In the introduction it is necessary to describe the acronym CDC, it is recommended that the first time an acronym is used it is described first. The same applies to the acronym LSTM in the "Literature Review" section. In addition, other acronyms that should be described are BMI, BMI.

Author response:  Thank you for your valuable comment. As per the reviewer’s suggestion, through out the entire article the acronyms are abbreviated.

2) It is noted that a literature review of works related to the proposal is presented; however, the main differences with respect to these works that allow identifying and highlighting the importance and novelty of this proposal are not indicated. In addition, it is suggested that this section be further enriched with a greater number of related works, preferably covering the last 7 years

Author response:  Thank you for your valuable comment. As per the reviewer’s suggestion, we have added additional references related to activity tracker.

3) It is suggested that after presenting a figure there should not immediately be a subtitle or section. This happens when presenting figures 1, 2, 3 and 12, so it is suggested to take care of this detail.

Author response:  Thank you for your comment. As per your valuable advice, we have rearranged the figures properly in the revised manuscript.

4) In the section "Hierarchical Agglomerative clustering" the formulas for Euclidean distance and Agglomerative clustering are presented, however, the wording does not refer to either formula.

5) Why Random Forest was used and not another classification algorithm?, it is suggested to justify this in the paper.

Author response:  Thank you for your comment. As per the valuable advice of the reviewer, the author has been updated the random forest and LSTM content with list of hyper parameters and its values in section 3. As the random forest consist of trees, in this current work, random forecast classification algorithm is compared with Decision Tree to justify the outperformance of ensemble of decision tress in random forest. According to this work, random forest is compared with ML methods like SVM and decision tree.

6) It is noted that in the section "Second level clustering" reference is made to figure 3, however figure 3 is presented in the previous section, so it is suggested that first reference be made to the figure and then the figure is presented. It is suggested to take care of this aspect with all the figures or tables presented in the document.

Author response:  Thank you for your comment. As per the valuable suggestion of the honorable reviewer, the contribution, thank you for your observation. We have rearranged  the figure at corresponding section.

7) In the Data Pre-processing, Figure 6 is presented, however, the figure is not referenced anywhere in the document.

Author response:  Thank you for your comment. As per the valuable suggestion of the honorable reviewer, the contribution, thank you for your observation. We have cited the figure at appropriate statement.

8) Make sure that the name of the figure 8 is on the same sheet as the figure.

Author response:  Thank you for your comment. As per the valuable suggestion of the honorable reviewer, the contribution, thank you for your observation. We have revised the figure caption and updated.

9) It is suggested that in the conclusions or in the discussion part, the main challenges or challenges identified with respect to people who exercise regularly or periodically should be described, but above all how to encourage people who are completely sedentary.

Author response:  Thank you for your input. As per the reviewer’s suggestion, included discussion about the work is included within the conclusion part section 5.

10) It is suggested to review and take care that the format required by the journal is complied with. (https://www.mdpi.com/journal/axioms/instructions )

Author response:  Thank you for your comment. As per the valuable suggestion of the honorable reviewer, the author have followed the template of the journal and formatted the revised manuscript.

Round 2

Reviewer 1 Report

The authors have attended to the concerns.

Author Response

Comment: The authors have attended to the concerns.

Author Response: Thank you very much for accepting our manuscript to publish in your esteemed journal.

Reviewer 2 Report

The proposal presents a dynamic physical activity recommendation delivered through a mobile fitness app, a deep learning approach. The proposal presents an actual and interesting topic; however, the following aspects were identified:

11) It is necessary to describe the acronym CDC (Centers for Disease Control and Prevention ?), LSTM (Long Short-Term Memory ?), BMI, SDG. It is recommended that the first time an acronym is used it is described first.

22) There is no reference to Table 3 throughout the document, however, on line 550 there is a reference to Table 1, this is a table numbering error ?.

33) The colon at the end of the title of section 5 should be eliminated “Conclusion and Future Plans”.

4.4) The following suggestion was not made or justified.

a.       It is suggested that in the conclusions or in the discussion part, the main challenges or challenges identified with respect to people who exercise regularly or periodically should be described, but above all how to encourage people who are completely sedentary.

Author Response

Thank you for your useful comments and suggestions to revise our manuscript. We have carefully revised the manuscript accordingly. The detailed revisions/corrections that have been undertaken and our detailed point by point response to all the comments given by the reviewers are listed below for your kind review.

 Comments and Suggestions for Authors

The proposal presents a dynamic physical activity recommendation delivered through a mobile fitness app, a deep learning approach. The proposal presents an actual and interesting topic; however, the following aspects were identified:

Comment 1: It is necessary to describe the acronym CDC (Centers for Disease Control and Prevention?), LSTM (Long Short-Term Memory ?), BMI, SDG. It is recommended that the first time an acronym is used it is described first.

Author response:  Thank you for your valuable comment. As per the reviewer’s suggestion, all the acronyms are abbreviated in the revised manuscript.

Comment 2: There is no reference to Table 3 throughout the document, however, on line 550 there is a reference to Table 1, this is a table numbering error?.

Author response:  Thank you for your comment. We have cited the tables with more care in the revised manuscript.

Comment 3: The colon at the end of the title of section 5 should be eliminated “Conclusion and Future Plans”.

Author response:  In Section 6(Conclusion and Future Plans), the colon at the end of the title have been removed.

Comment 4: It is suggested that in the conclusions or in the discussion part, the main challenges or challenges identified with respect to people who exercise regularly or periodically should be described, but above all how to encourage people who are completely sedentary.

Author response:  Thank you for your valuable comment. As a new section (section 5), discussion part has been included with main challenges in the revised manuscript.